# *NtMYB3*, an R2R3-MYB from Narcissus, Regulates Flavonoid Biosynthesis

**DOI:** 10.3390/ijms20215456

**Published:** 2019-11-01

**Authors:** Muhammad Anwar, Weijun Yu, Hong Yao, Ping Zhou, Andrew C. Allan, Lihui Zeng

**Affiliations:** 1College of Horticulture, Fujian Agriculture and Forestry University, Fuzhou 35002, China; anwar_uaar@yahoo.com (M.A.);; 2The New Zealand Institute for Plant & Food Research, Mt Albert Research Centre, Private Bag 92169, Auckland 1025, New Zealand; andrew.allan@plantandfood.co.nz; 3School of Biological Sciences, University of Auckland, Private Bag 92019, Auckland 1142, New Zealand

**Keywords:** Chinese narcissus, flavonol, R2R3-MYB, flavonoid repressor, anthocyanin

## Abstract

R2R3-MYB transcription factors play important roles in the regulation of plant flavonoid metabolites. In the current study, *NtMYB3*, a novel R2R3-MYB transcriptional factor isolated from Chinese narcissus (*Narcissus tazetta* L. var. *chinensis*), was functionally characterized. Phylogenetic analysis indicated that NtMYB3 belongs to the AtMYB4-like clade, which includes repressor MYBs involved in the regulation of flavonoid biosynthesis. Transient assays showed that *NtMYB3* significantly reduced red pigmentation induced by the potato anthocyanin activator *StMYB-AN1* in agro-infiltrated leaves of tobacco. Over-expression of *NtMYB3* decreased the red color of transgenic tobacco flowers, with qRT-PCR analysis showing that *NtMYB3* repressed the expression levels of genes involved in anthocyanin and flavonol biosynthesis. However, the proanthocyanin content in flowers of transgenic tobacco increased as compared to wild type. *NtMYB3* showed expression in all examined narcissus tissues; the expression level in basal plates of the bulb was highest. A 968 bp promoter fragment of narcissus *FLS* (*NtFLS*) was cloned, and transient expression and dual luciferase assays showed *NtMYB3* repressed the promoter activity. These results reveal that *NtMYB3* is involved in the regulation of flavonoid biosynthesis in narcissus by repressing the biosynthesis of flavonols, and this leads to proanthocyanin accumulation in the basal plate of narcissus.

## 1. Introduction

Chinese narcissus (*Narcissus tazetta* L. var. *chinensis* Roem) belongs to the Amaryllidaceae family and is a perennial bulbous plant widely cultivated in East Asia and China. It has popular ornamental flowers and fragrance with significant cultural and commercial value. Chinese narcissus has been reported as a triploid species, generally propagated by vegetative methods. Traditional cross breeding is complicated [1]. Therefore, only a limited number of cultivars have been exploited for commercial production in China, and agronomic characters, such as flower colors, have yet to be improved [2].

Flower color is a very important consideration for customer choice. The pigments that color most flowers are commonly flavonoids. Flavonoids belong to a class of secondary metabolites, which are mainly classified into main subgroups, including isoflavones, flavones, flavonols, flavan-diols, chalcones, anthocyanins, and proanthocyanidins (condensed tannins) [3]. Anthocyanins constitute a large subclass of flavonoids and are natural plant pigments that give diverse colors—typically purple, red, and blue—to flowers, fruits, and vegetables [4,5]. Proanthocyanidins (PAs) derived from the condensation of flavan-3-ols units such as catechin, epicatechins, and epigallocatechin are one of the major groups of polyphenolic compounds produced through the flavonoid biosynthetic pathway [6]. Proanthocyanidin broadly exists in the plant kingdom [7]. They are found in flowers, seeds, bark, and leaves and play a key role in defense in response to plant diseases. PAs improve health benefits and quality of plant products including fruits and wine [8].

Flavonoids are also synthesized through the phenylpropanoid pathway. Phenylalanine acts as the precursor molecule for flavonoid biosynthesis, which is transformed to cinnamic acid by phenylalanine ammonia lyase (PAL). One molecule of CoA-ester of cinnamic acid and three molecules of malonyl-CoA are reduced into the naringenin chalcone, and the reaction is catalyzed by chalcone synthase (CHS). The chalcone is isomerised to a flavanone by chalcone flavanone isomerase (CHI). The biosynthesis of the different branches of flavonoids including anthocyanins, proanthocyanidins and flavonols shares general precursors. Dihydroflavonols act as substrates for dihydroflavonol 4-reductase (DFR) or flavonol synthase (FLS), and therefore, these various metabolite branches compete with each other, resulting in flavonol, anthocyanin, or PA compositions of secondary metabolites.

Flavonoid biosynthesis is modulated by various transcriptional factors (TFs) belonging to diverse families such as MYB, basic helix-loop-helix (bHLH), and WD40 proteins, which constitute a MYB-bHLH-WD40(MBW) complex [9]. MYB proteins play a key role in controlling target genes in the flavonoid biosynthetic pathway.

The regulation of the flavonoid biosynthetic pathway in dicotyledons species occurs by diverse patterns. Specific MYB transcriptional factors have been proposed to control branches of the flavonoid biosynthetic pathway in plants. For instance, AtMYB11, AtMYB12, and AtMYB111 are responsible for the biosynthesis of flavonol in Arabidopsis, whereas AtPAP1 controls the anthocyanin biosynthesis pathway. *AtMYB75*, *AtMYB90*, *AtMYB113*, and *AtMYB114* are also involved in the regulation of the anthocyanin biosynthetic pathway [9,10,11,12]. Other MYB transcriptional factors simultaneously control many branches of flavonoid biosynthesis. For example, in grape, VvMYB5a and VvMYB5b are known to regulate the biosynthesis of several branches, including anthocyanins, PAs, flavonols, and lignins [13]. In monocotyledon plants, most *R2R3-MYB* genes reported in flavonoid biosynthetic pathway regulate anthocyanin pigmentation, such as *ZmC1* and *ZmPl* in maize, *OgMYB1* in orchid, and *LhMYB6*/*LhMYB12* in Asiatic Hybrid Lily [14,15].

In addition to the activator MYBs, some repressors of the flavonoid pathway, particularly anthocyanins, have been reported, including an R2R3-MYB repressor from strawberry FaMYB1 [16], Arabidopsis AtMYB4 [17], and a single domain MYB in Arabidopsis, AtMYBL2 [18]. NtMYB2, a R2R3-MYB in Chinese narcissus, was found to repress anthocyanin biosynthesis [19]. Two R3-MYB repressors of anthocyanin, LhR3MYB1 and LhR3MYB2, were identified in lily [20]. However, other negative regulators of flavonoids in monocot plants still remain to be studied.

In Chinese narcissus, flower color is either yellow or white. The composition of flavonoid compounds in flowers of narcissus cultivars was examined, with flavonols detected and no anthocyanins found [19]. Previous studies showed that PAs are accumulated in the scales, roots, and basal plates of Chinese narcissus, but not in flowers. A high expression level of the genes that encode the *FLS* or *DFR* steps play a key role in regulating flux through the different branches of the flavonoid biosynthetic pathway [21]. Understanding the regulation of different branches in flavonoid biosynthesis will provide knowledge to produce anthocyanins by genetic engineering in narcissus. In the current investigation, a novel R2R3-MYB repressor NtMYB3 was isolated from Chinese narcissus and functionally characterized. The results suggest that NtMYB3 may be a repressor of flavonol biosynthesis. The putative role of NtMYB3 provides an understanding of the regulatory mechanisms of flavonoid biosynthesis in Chinese narcissus as well as other monocot species.

## 2. Results

### 2.1. Cloning and Sequence Analysis of NtMYB3

Sequence analysis showed that NtMYB3 has a full length ORF of 603bp. *NtMYB3* encodes a putative protein of 201 amino acids with a molecular weight of 23 kDa, which has 83% amino acid sequence identity with Allium *cepa* AcMYB1 and 78% with *Vitis vinifera* VvMYB4a which are both MYB repressors. Insilico investigation (http://nls-mapper.iab.keio.ac.jp/cgi-bin/NLS_Mapper_form.cgi) for subcellular localization showed the occurrence of nuclear localization signal “RPDLKRGNFTEEEDDLIIKLHSL” starting from the amino acid at 62nd location. Results of phylogenetic tree analysis indicated that amino acid sequences of R2R3-MYB repressor genes from various plants were divided into two clades (Figure 1). Clade 1 included related MYB transcriptional factors grouped with the AtMYB4-like clade. Clade 2 was composed of associated MYB transcriptional factors grouped with the FaMYB1-like clade. Both of clades transcriptional factors contained the C2 repressor motifs. NtMYB3 was grouped with AcMYB1, *Arabidopsis* AtMYB4, AtMYB32, and VvMYB4a, belonging to subgroup 4 of the R2R3-MYB transcriptional factors. NtMYB2, which was previous functionally characterized as a repressor of anthocyanin, belongs to same clade with NtMYB3 [19] (Figure 1).

Protein sequence alignment between NtMYB3 and numerous R2R3-MYB repressors showed that NtMYB3 consisted of conserved R2 and R3 DNA binding domains at the N-terminus. R3 DNA binding domains of NtMYB3 contained a bHLH binding motif [D/E]Lx_2_[R/K]x_3_Lx_6_Lx_3_R, as previously identified [22]. The conserved motif “DNEI” was found in R3 DNA binding domains of NtMYB3. In addition to R2R3 binding domains, NtMYB3 have two typically conserved signature or motifs C1 (KLIsrGIDPxT/SHRxI/L) and C2 (pdLNLD/ELxiG/S) found at the C-terminus, as previously identified in subgroup 4 R2R3-MYB transcriptional factors. On the other hand, the C-terminal downstream conserved motifs indicate high divergence (Figure 2).

*Arabidopsis thaliana* AtMYB32(NP 195225), *Fragaria ananassa* FaMYB1 (AAK84064.1), *Petunia hybrid* PhMYB27 (AHX24372.1), *Vitis vinifera* VvMYBC2-L1 (ABW34393), *Vitis vinifera* VvMYBC2_L3 (AIP98385.1), *Gossypium hirsutum* GhMYB6 (AAN28286), *Populus tremuloides* PtrMYB182 (Potri.004G088100), *Arabidopsis thaliana* AtMYB4 (NP_195574.1), *Vitis vinifera* VvMYB4a (ABL61515.1), *Antirrhinum majus* AmMYB308 (P81393), *Glycine max* GmMYB54 (ABHO2822.1), *Malus domestica* MdMYB16 (HM122617.1), *Punica granatum* PgMYB1 (AJD79907.1), *Salvia miltiorrhiza* SmMYB39 (KC213793)*, Narcissus tazetta* NtMYB3 (KC763975.1), NtMYB2 (KY860527.1), *Vitis vinifera* VvMYBC2-L2 (ACX50288).

*Arabidopsis thaliana* AtMYB32(NP 195225), *Fragaria ananassa* FaMYB1 (AAK84064.1), *Petunia hybrid* PhMYB27 (AHX24372.1), *Vitis vinifera* VvMYBC2-L1 (ABW34393), *Vitis vinifera* VvMYBC2-L3 (AIP98385.1), *Gossypium hirsutum* GhMYB6 (AAN28286), *Populus tremuloides* PtrMYB182 (Potri.004G088100), *Arabidopsis thaliana* AtMYB4 (NP_195574.1), *Vitis vinifera* VvMYB4a (ABL61515.1), *Antirrhinum majus* AmMYB308 (P81393), *Glycine max* GmMYB54 (ABHO2822.1), *Malus domestica* MdMYB16 (HM122617.1), *Punica granatum* PgMYB1 (AJD79907.1), *Salvia miltiorrhiza* SmMYB39 (KC213793)*, Narcissus tazetta* NtMYB3 (KC763975.1), NtMYB2 (KY860527.1), *Vitis vinifera* VvMYBC2-L2 (ACX50288).

### 2.2. Transient Expression of NtMYB3 in Tobacco

To check the function of *NtMYB3*, transient assays were performed in tobacco. Noticeable red pigmentation was found 3–4 days after infiltration with *StMYB-AN1*, which is a potato anthocyanin activator [23]. No anthocyanin accumulation or red pigmentation was found with infiltration of *NtMYB3*. Red pigmentation or anthocyanin accumulation significantly decreased in leaves of tobacco when co-infiltrated with *NtMYB3* and *StMYB-AN1*. Furthermore, anthocyanin accumulation was also significantly reduced in leaves co-infiltrated with *NtMYB3*, *StMYB-AN1*, and *StbHLH* (Figure 3).

### 2.3. The Over-Expression of NtMYB3 Decreases the Flower Pigmentation of Transgenic Tobacco

*NtMYB3*, under control of the 35S promoter, was transferred into tobacco by leaf disc transformation using *Agrobacterium*. Five transgenic lines carrying *NtMYB3* gene were obtained. The transgenic plants were confirmed by PCR. Transgenic flowers carrying *NtMYB3* genes displayed a significant decrease in color as compared to control, which generate pink flowers, changing from light pink to almost white. One *NtMYB3* over-expression line (L-2) showed almost white flowers with light red veins at the tips of petals. Transgenic lines (L-4 and L-5) showed very weak red flowers (Figure 4A). Moreover, in transgenic lines, the positions of stigmas were above the anther, as compared to wild type, because of the style extending (Figure 4B). Furthermore, anthocyanin contents of wild type and transgenic flowers were analysed. The anthocyanin content of flowers carrying *NtMYB3* gene was significantly decreased compared to those of wild type (Figure 5).

### 2.4. Over-Expression of NtMYB3 Affects Expression Level of Flavonoid Key Genes in Transgenic Flowers of Tobacco

To check the effects of ectopic over-expression of *NtMYB3* on biosynthetic genes involved in the flavonoid biosynthesis pathway in transformed tobacco flowers, qRT-PCR analysis was performed. The results of qPCR analysis indicated that over-expression of *NtMYB3* strongly affected the transcript level of the biosynthetic genes involved in flavonol, anthocyanin, and proanthocyanin biosynthesis pathways. Expression of *CHS*, *CHI, F3H, DFR*, *ANS*, and *UFGT*, were significantly down-regulated in transgenic flowers (Figure 6), especially the *UFGT* gene. In addition, the expression level of *FLS* gene encoding for flavonol biosynthesis was significantly decreased in transgenic lines with respect to wild type. However, PA biosynthetic pathway genes *LAR* and *ANR* were up-regulated and had higher expression levels in transgenic flowers of all *NtMYB3* over expression lines (Figure 6). To confirm the up-regulation of *LAR* and *ANR* genes and the down-regulation of *FLS*, PA and flavonol contents of wild type and transgenic flowers were calculated. The PA contents in transgenic flowers carrying *NtMYB3* gene were significantly increased, while the flavonol contents decreased as compared to wild type (Figure 7).

### 2.5. The Expression Pattern of NtMYB3 Gene in Different Tissues of Chinese Narcissus

QRT-PCR was carried out to determine the expression levels of *NtMYB3* in various tissues of Chinese narcissus. These results showed that *NtMYB3* was expressed in petals, corona, leave, scale, and basal plates of narcissus (Figure 8). The expression level of *NtMYB3* in basal plates was highest, as compared to leaves, bulb scales, petals, and the corona (Figure 9). These results suggested that *NtMYB3* may play a role in the flavonoid biosynthesis of basal plates.

### 2.6. Regulation Analysis of NtMYB3 to the NtFLS Promoter

A 968 bp promoter fragment of narcissus *NtFLS* was amplified from genomic DNA by genome walking (GenBank number MH 472580). The promoter contained 12 TATA-box, 10 CAAT-box elements, and other promoter elements in the promoter region (Appendix A). Two MYB transcription factor elements were found within the promoter sequence. In addition, the light-responsive elements, including Box I, G-box, G-Box, boxII, Sp1, TCT-motif, 3-AF1binding site, and ACE, were found.

Transient expression in tobacco leaves was used to investigate whether NtMYB3 regulates the cloned *NtFLS* promoter. A histochemical GUS assay showed that NtMYB3 decreased pNtFLS::GUS activity (Figure 10A,B). The expression level of *GUS* was decreased by 58% (Figure 10C).

A dual luciferase assay was carried out to verify the interaction between NtMYB3 and *NtFLS* promoter. NtMYB4, a flower-specific MYB homologous to Arabidopsis MYB21 that regulates stamen development and pollen phenylpropanoid [24], was found to elevate *NtFLS* levels (unpublished) and used in the assay. The results obtained with the GUS activity assay results, indicating that NtMYB3 significantly decreased the activity of *NtFLS* promoter under conditions of activation by NtMYB4 (Figure 11).

*Agrobacterium* cultures containing the reporter construct pNtFLS:LUC were mixed with the effector constructs GUS (empty), NtMYB3, NtMYB4, and NtMYB3+NtMYB4 separately and co-infiltrated into *N. benthamiana* leaves. Small letter showed the significant difference at the level of *p* < 0.05 by using LSD test. The error bars stand for the SE of three biological replicates.

## 3. Discussion

### 3.1. The Sequence of NtMYB3 Has Features of an R2R3-MYB Repressor

The C2 repressor motif is a general feature of EAR-type transcriptional repressors, with changes in the amino acid residues within the EAR motif resulting in a decrease or loss of suppression. In Arabidopsis, inhibitory activity was lost by replacing amino acid residues D with A in the EAR motif [25]. In apple, MdMYB16 has an EAR repressor motif at the C-terminus (PDLNLDLQIS) when removed, suppresses the inhibition of anthocyanin [26]. Sequence analysis showed that NtMYB3 had this C2 motif, suggesting NtMYB3 may have a repressor function.

Phylogenetic tree analysis revealed that repressor MYB transcriptional factors were separated into AtMYB4-like and FaMYB1-like clades. In the present study, NtMYB3 is related to the AtMYB4-like clade, which also includes NtMYB2 [19]. The R2R3-MYB members of the AtMYB4-like clade were previously functionally characterized and are involved in the down-regulation of the flavonoid biosynthetic pathway. Therefore, NtMYB3 may be a repressor of flavonoid biosynthesis.

### 3.2. NtMYB3 Regulates Flavonoid Biosynthesis in Tobacco

In potato, *StMYB-AN1* positively regulates anthocyanin biosynthesis [23]. Visible red pigmentation was induced in agro-infiltrated tobacco leaves with individual *StMYB-AN1*. A significantly reduction of red pigmentation or patches was found when young leaves were co-injected with *StMYB-AN1* and *NtMYB3*. Ectopic over-expression of *NtMYB3* in tobacco decreased petal pigmentation, and total anthocyanin was reduced in transgenic tobacco flowers, suggesting that NtMYB3 is an anthocyanin repressor in tobacco. R2R3-MYB repressors have two kinds, one of which functioned on MBW complexes such as FaMYB1, and the other directly binds on target genes such as AtMYB4 [26]. In the presence of StbHLH co-infiltration, the red pigment was reduced as only co-injected with *StMYB-AN1* and *NtMYB3*, indicating that NtMYB3 may not compete for StbHLH with StMYB and may play a role by directly binding with the promoters of structural genes.

Our study is consistent with the previous results of R2R3-MYB repressors in dicotyledons. Ectopic expression of *VvMYB4* in tobacco decreased anthocyanin accumulation in flowers [27]. Poplar *PtrMYB57* decreases the content of anthocyanin in transgenic poplar plants as compared to the control [28]. Similar results were also obtained in our previous study, which showed that another R2R3-MYB in Chinese narcissus, *NtMYB2*, repressed the anthocyanin accumulation in tobacco transgenic flowers [19].

In dicotyledons, R2R3-MYBs are reported to repress expression of genes of the flavonoid biosynthesis pathway. The main effect of *FaMYB1* overexpression is at the lower end of the flavonoid biosynthetic pathway, more directly related to the biosynthesis of anthocyanins and the flavonol quercetin [16]. Overexpression of *AtMYB4* in Arabidopsis affects the up-stream flavonoid biosynthesis genes such as *C4H*, *CHS* and *4CL* [17]. In grapevine, *VvMYB4-like* and *VvMYB4A* decreased anthocyanin accumulation by suppressing the expression of the *DFR*, *ANS* and *UFGT* genes respectively [27,29]. In *NtMYB3* tobacco flowers, nearly all structural genes involved in anthocyanin and flavonol biosynthetic pathway were down-regulated. The results suggest NtMYB3 regulates anthocyanin and flavonol biosynthesis in tobacco, acting as a transcriptional repressor of flavonoid structural genes. However, the expression of *ANR* and *LAR* was up-regulated, and PA contents increased in transgenic flowers, in contrast to the suppression of anthocyanin and flavonol biosynthesis. This observation requires further study.

In addition to changes in floral color pattern, changes in other morphological traits were seen in *NtMYB3* lines. Pistil length of over-expression *NtMYB3* lines was extended compared to wild type, with stigmas above the anther in transgenic flowers. Similar observations were reported for *AmMYB308*, which when over expressed in transgenic tobacco enhanced styles of the pistil [30]. However, the mechanisms of how NtMYB3 can induce morphological character changes require further study.

### 3.3. NtMYB3 Suppresses the Transcription of NtFLS in Chinese Narcissus

Previous evidence has shown that perianths and coronas of Chinese narcissus have a higher level of flavonols, PAs are accumulated in basal plates, and no anthocyanidins are detected in all organs [21]. Expression analysis indicated that *NtMYB3* has a higher expression level in basal plates than petals and corona, which contrasts with *NtMYB2*, which has higher expression levels in petals and corona [19]. Previous studies have suggested that decreased expression of *NtFLS* and a lack of *ANS* expression lead to PAs accumulation in basal plates of Chinese narcissus [21]. Therefore, it is suggested that NtMYB3 may play a repressive role in basal plates of Chinese narcissus by down-regulating *NtFLS* expression. Results of transient expression in tobacco leaves and dual luciferase assays verify the regulation of the promoter of *NtFLS* by NtMYB3. Our results also indicate the different roles between NtMYB3 and NtMYB2 in Chinese narcissus, although both are AtMYB4-like repressors.

## 4. Materials and Methods

### 4.1. Plant Materials

Chinese narcissus (*Narcissus tazetta* L. var. *chinensis*) “Jin Zhan Yin Tai” was used in this experiments. The samples of petals, corona, leaves, bulb scales, and basal plates of Chinese narcissus were collected and further used for the RNA extraction.

Tobacco (*Nicotiana tabacum*) was used in transient expression and stable transformation. The young and healthy leaves of *Nicotiana benthamiana* were used for dual luciferase assay. The seeds of tobacco were sterilized with 75% alcohol and followed by 10% sodium hypochlorite (NaClO), then washed 3–5 times with dH_2_O, and cultured on MS medium. Leaves of sterile plantlets are used for stable transformation. The transgenic flowers of tobacco were collected and used for the extraction and estimation of total anthocyanins and PAs.

### 4.2. Cloning and Sequence Analysis of NtMYB3

The full length ORF of *NtMYB3* was obtained by designing gene-specific primers according to the published sequence (KC763975.1) (Appendix A). The molecular weight, isoelectric point (pI), and theoretical *NtMYB3* coding sequence translation was predicted using an online ExPASy server (www.ExPASy.org). The protein sequence of R2R3-MYB transcriptional factors from other various species were collected from NCBI (www.ncbi.nlm.nih.gov/BLAST/). The amino acid sequence of *NtMYB3* and other crops were aligned by using DNAMAN6.0 software with default parameters (Lynnon Corporation, San Ramon, CA, USA). The whole protein sequence of R2R3-MYB transcriptional factors related to NtMYB3 were used to develop a phylogenetic tree using MEGA5 software with default parameters. Bootstrap was 1000.

### 4.3. Expression Vector Construction

Plant expression vector was constructed using a cloning Kit (In-Fusion^®^HD, Clonetech, Palo Alto, CA, USA). The entire ORF of *NtMYB3* was obtained by primers with special restriction enzyme site. The forward primer was added with an *Eco*RI restriction site and reverse primer with a *Hin*dIII restriction site (Appendix A). The sequence was then cloned into a plant expression vector pSAK277 with the CaMV35S promoter. The constructed expression vector pSAK277-*NtMYB3* was transformed into the *Agrobacterium tumefaciens* strain GV3101 (MP90) by electroporation. A positive colony was chosen on medium (YEB) supplemented with 100 mg/L rifamycin (Solarbio, Beijing, China) and 50 mg/L kanamycin ((Solarbio, Beijing, China)) for transient and stable genetic transformation of tobacco.

### 4.4. Q-PCR

The expression of the *NtMYB3* gene in different tissues of Chinese narcissus was analyzed by real time qRT-PCR. The total RNA from various tissues was extracted using Omega Total RNA Kit (Omega, Norcross, GA, USA). cDNA was synthesized after removal of genomic DNA usinga PrimeScript^TM^ RT reagent Kit (Takara, China). QPCR was carried out with Light cycler^®^ 480 real time PCR (Roche Diagnostics, Indianapolis, IN, USA). The qRT-PCR condition was same as in our previous study [19]. The comparative transcript level was calculated with the Chinese narcissus *Actin* gene (GenBank number JN204912.1). The comparative Ct technique was performed to calculate the gene transcript level [31]. Three technical and biological replications for each sample were performed.

### 4.5. Transient Assays to Investigate NtMYB3 Function

For transient assays, the vectors pSAK277-*NtMYB3*, pSAK277-*StbHLH*, pSAK277-*StMYB-AN1*, and empty vector pSAK277 (as a negative control) were introduced into *Agrobacterium* stain GV3101 (MP90). *StMYB-AN1* and *StbHLH* were provided by Dr. Liu [23]. Transient assays were performed by the method, as previously described [19]. *NtMYB3, StMYB-AN1*, *NtMYB3+StMYB-AN1*, and *StMYB-AN1+NtMYB3+StbHLH* were carried out as treatments separately. *StMYB-AN1*+pCAMBIA1301 were used as a control treatment. Each treatment was replicated three times in transient assays. After 4–5 days of agro-infiltration injection, changes in color pigmentation were noticed in leaves. The collected samples were stored at −80 °C refrigerator for further study.

### 4.6. Stable Transformation of Tobacco

Stable transformation of tobacco (*Nicotiana tabacum*) through a leaf disc method was performed as previously reported [32]. The putative transformed shoots were selected on MS basal medium with the addition of 350 mg/L carbenicillin and 100mg/L kanamycin. In order to confirm the transgenic tobacco plants, PCR assay was carried out. The transgenic plants were moved to soil for their normal growth.

### 4.7. Extraction and Quantification of Proanthocyanins, Flavonol and Total Anthocyanins

Total contents of anthocyanin were extracted and quantified in transgenic flowers of tobacco using the method as previously described [33]. For the analysis of proanthocyanins, extraction and estimation of PAs was carried out as previously described [34]. The total flavonol was extracted and quantified in transgenic flowers of tobacco according to the method as previously described [35].

### 4.8. Isolation and Sequence Analysis of NtFLS Promoter

Genomic DNA was isolated from the leaves of Chinese narcissus using the CTAB method. The *NtFLS* promoter was isolated using the Genome Walking Kit (TaKaRa, Dalian, China) following the manufacturer’s instructions. The nested PCR analysis was carried out with three sets of primers, as well as three gene-specific primers, which were NtFLS-R1:5′-ATGAAACAAGTAGTCGACCCAAGC-3′, NtDFR-R2: 5′-TCACGTAAGCCTCCTTCTCCCCTTGT-3′, and NtDFR-R3: 5′-TGGTTCACGATTTGGAAGATCCCC-3′. Putative *cis*-acting elements of the promoter were predicted using the PLACE databases (http://www.dna.affrc.go.jp/PLACE/signalscan.html) and Plant CARE (http://bioinformatics.psb.ugent.be/webtools/plantcare/html/).

### 4.9. Transient Expression to Investigate the Regulation of NtMYB3 to NtFLS Promoters

The promoter of *NtFLS* was constructed into the plant expression vector pBI121, replacing the 35S promoter before GUS gene. The recombinant vector pBI121-pNtFLS::GUS was transformed into the *Agrobacterium tumifaciens* GV3101(MP90), and their agro-infiltration suspension (10 mM MgCl_2_, 10 mM 2-(N-Morpholino) ethanesulfonic acid (MES), 200 μM Acetosyringone incubated for one hour at 28 °C) was injected into the young leaves of tobacco was mixed with pSAK277-*NtMYB3* at the ratio of 1:1 as described by NayeriandAnbuhi [36]. In this assay, three biological replications were performed.

After three days, tobacco leaves were collected and RNA was isolated, real-time qPCR was performed using TaKaRa SYBR Premix ExTaq^TM^II (Takara, Dalian, China) to analyze the expression levels of *GUS* gene of various treatments. Primers of GUS gene are GUS-F: TACCGTACCTCGCATTACCC, GUS-R: CTGTAAGTGCGCTTGCTGAG. At the same time, histochemical GUS assay was performed to measure GUS activity. Tobacco leaf tissues were incubated in GUS staining solution (2 mM 5-bromo-4-chloro-3-indolyl-b-D-glucuronide acid in 0.1 M sodium phosphate pH 7.0, 0.1% (g/mL) Triton X-100, 0.5 mM K_3_FeCN_6_, 0.5 mM K_4_FeCN_6_, 10 mM Na_2_EDTA) for one day at 37 °C. After staining, the chlorophyll was liberated by extraction in 100% ethanol for 1 h and 75% ethanol for 1 h.

### 4.10. Dual Luciferase Assay

The dual luciferase test was performed as previously described by Espley, et al. [36]. The promoter of *NtFLS* gene cloned in pGreenII 0800-LUC was used as the reporter cassette, TFs *NtMYB3*, and *NtMYB4* in pSAK277 driven by the cauliflower mosaic virus (CaMV) 35S promoter as the effector cassette, and the negative control β-glucuronidase (GUS) in pSAK277 was used in assays. *Agrobacterium* cultures (10 mM MgCl_2_, 10 mM 2-(N-Morpholino) ethanesulfonic acid (MES), 200 μM Acetosyringone incubated for one hour at 28 °C) containing the reporter cassette and the effector cassettes were mixed and co-infiltrated into *N. benthamiana* leaves as previously described by NayeriandAnbuhi [37]. Plants were placed in growth chamber for three days. For each treatment, the samples of leaf discs about 1 cm in diameter were taken. Cell lysate was put into the grinded sample and then followed by incubation for 5 min on ice. Firefly luciferase (LUC) values are reported relative to the Renilla luciferase (REN) control. The assays were carried out according to the instruction of dual luciferase reporter assay kit (Yeasen Biotech Co, Ltd. Shanghai, China) using the Luciferase Assay System (Tecan infinite M200 PRO, (Yeasen Biotech Co, Ltd. Shanghai, China), and the setting for fluorescence value measurement reading time was 1000 ms in the luminometer.

### 4.11. Statistical Analysis

In this study, statistical analysis was performed by one-way ANOVA. LSD values were estimated using *p* = 0.05 and used to compare treatment means.

## Figures and Tables

**Figure 1 ijms-20-05456-f001:**
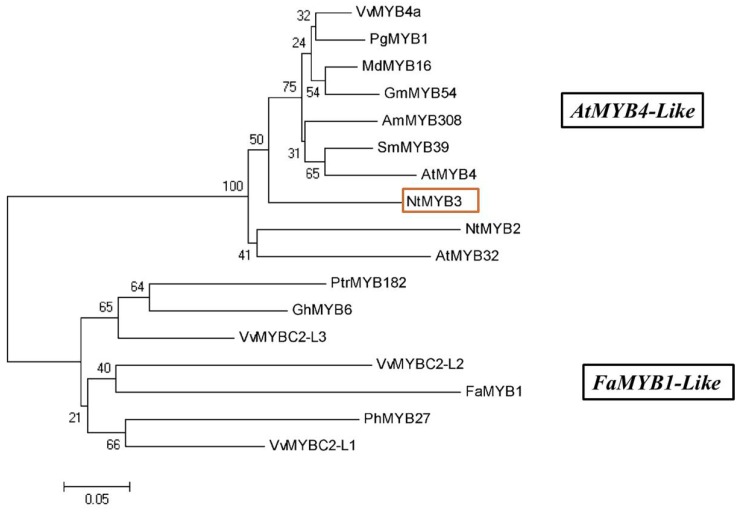
Phylogenetic tree analyses of MYB repressors. Our candidate gene NtMYB3 are shown by red rectangle.

**Figure 2 ijms-20-05456-f002:**
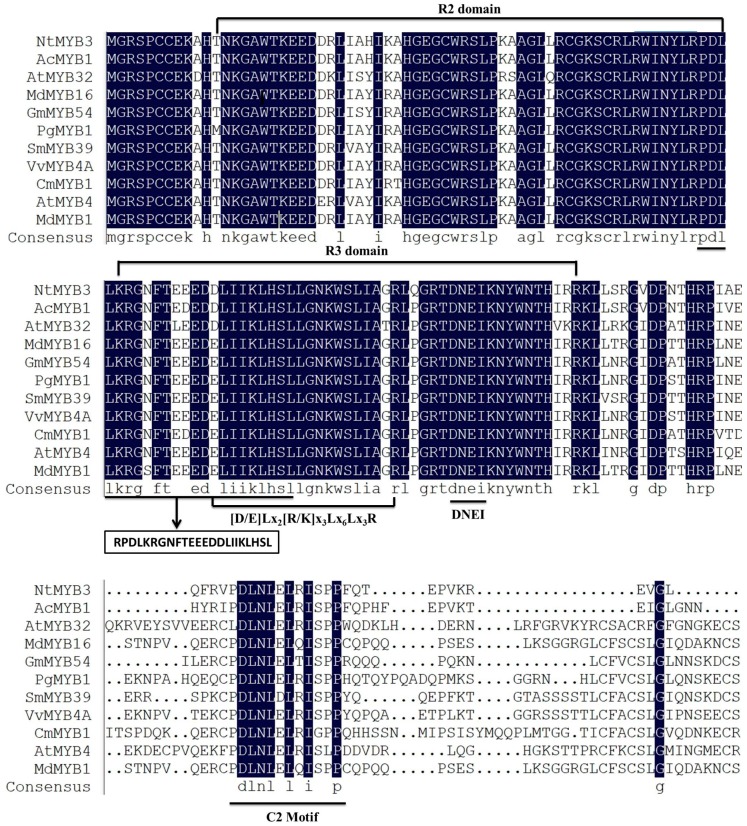
Multiple alignments of the protein sequences of NtMYB3 with other R2R3-MYB repressors belonging to subgroup 4. Blue shaded are indicated the 100 % homology suquences of amino acids.

**Figure 3 ijms-20-05456-f003:**
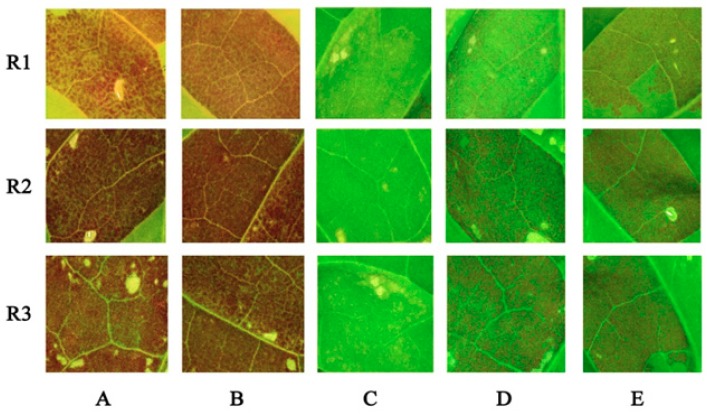
Transient transformation of tobacco leaves (*Nicotiana tabacum*). R1, R2, and R3 represent three replicates. (**A**) Infiltration with *StMYB-AN1*; (**B**) Co-infiltration of pSAK277 with *StMYB-AN1*; (**C**) Infiltration with *NtMYB3*; (**D**) Co-infiltration of *StMYB-AN1*with *NtMYB3*; (**E**) Co-infiltration of *StMYB-AN1*, *NtMYB3*, and *StbHLH*. Photos were taken after five days of agro-infiltration injection. (Scale bar = 1 cm).

**Figure 4 ijms-20-05456-f004:**
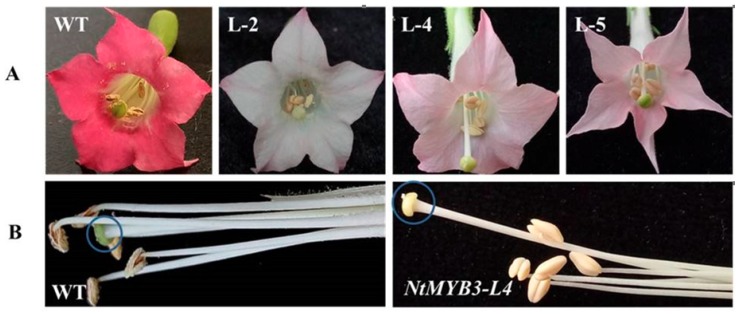
Floral phenotypes of transgenic tobacco plants carrying *NtMYB3* gene.Wild type tobacco (WT) and transgenic lines of tobacco are (L-2, L-4 and L-5). Flower color comparison in wild type (WT) and transgenic lines (**A**); Comparison of pistil length in WT and transgenic tobacco (**B**). Blue circle indicated the top of the pistal length.

**Figure 5 ijms-20-05456-f005:**
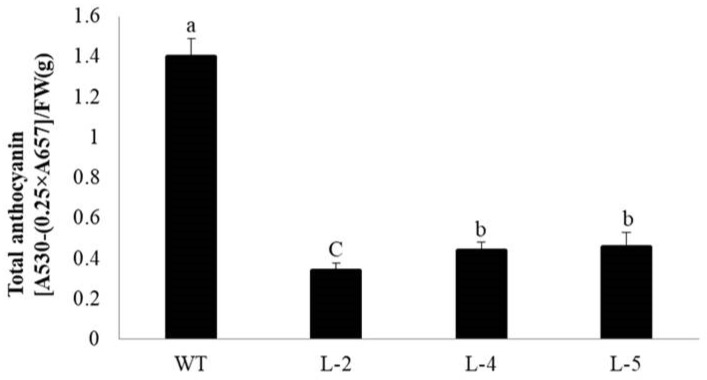
Estimation of total anthocyanin in wild type and transgenic lines (L-2, L-4, and L-5). The bars represent the standard error of thrice biological replication. Small letters showed a significant difference at the level of *p* < 0.05 by using LSD statistical analysis.

**Figure 6 ijms-20-05456-f006:**
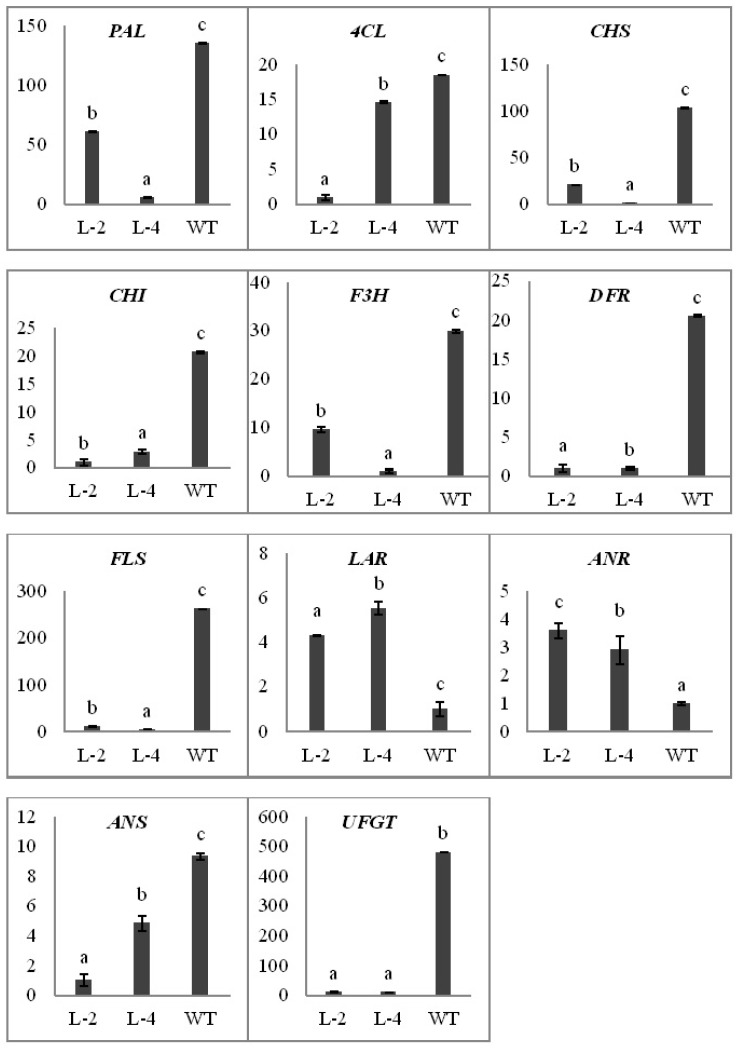
Expression analyses of flavonoid biosynthetic pathway key genes in transgenic flowers carrying *NtMYB3* by qPCR. L-2, L-4, indicated two transgenic tobacco lines. PAL, Phenylalanine ammonia lyase; 4CL, 4-coumaroyl-CoA ligase, CHS, Chalcone synthase; CHI, Chalcone isomerase; F3H, Flavanone 3-hydroxylase; DFR, Dihydroflavonol 4-reductase; FLS, flavonol synthase LAR, leucoanthocyanidin reductase; ANR, anthocyanidin reductase; ANS, anthocyanin synthase; UFGT, UDP glucose-flavonoid 3-O glucosyltransferase. Small letter showed the significant difference from wild type at the level of *p* < 0.05 by using LSD test. The bar represents the standard error of thrice biological replications.

**Figure 7 ijms-20-05456-f007:**
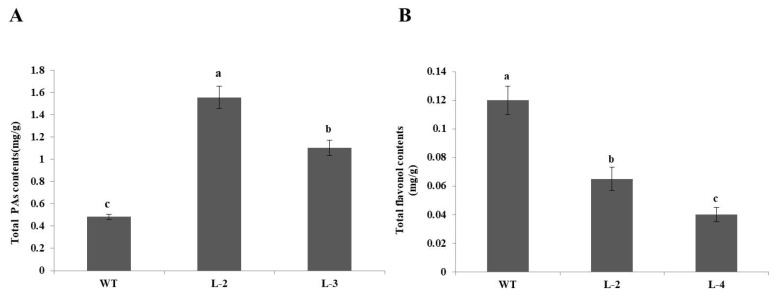
Quantification of proanthocyanin and flavonol in transgenic and wild type tobacco flowers. (**A**) PA contents and (**B**) flavonol contents. Small letters showed a significant difference at the level of *p* < 0.05 by using LSD statistical analysis.

**Figure 8 ijms-20-05456-f008:**
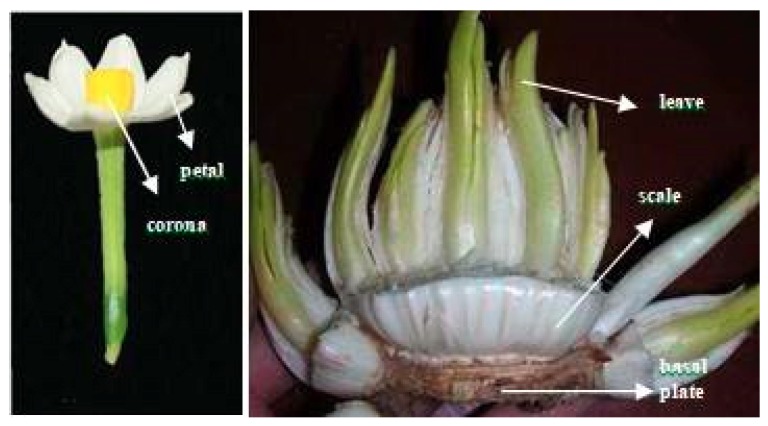
Different tissues and organs of Chinese narcissus used in this experiment.

**Figure 9 ijms-20-05456-f009:**
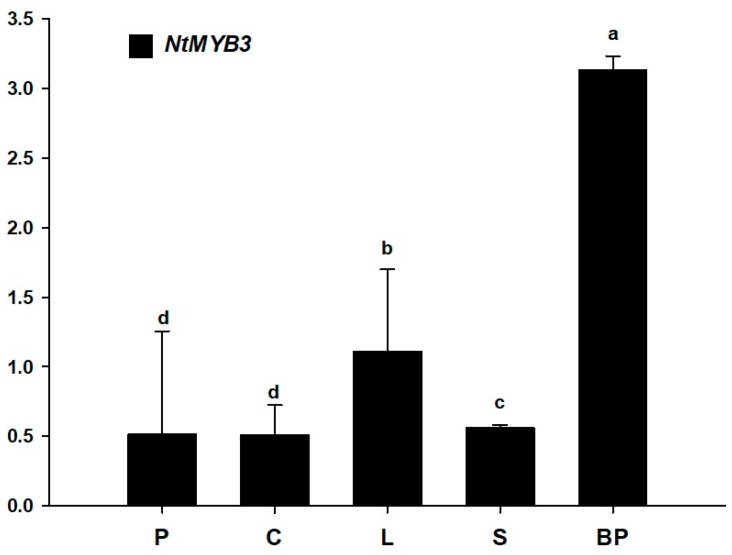
Expression analysis of *NtMYB3* in various tissues of *Narcissus tazetta*. P, C, L, S, and BP indicate the petals, corona, leaves, bulb scales, and basal plates, respectively. Small letters showed the significant difference among different tissues at the level of *p* < 0.05 by using LSD test. Bars showed the standard error of three replicates.

**Figure 10 ijms-20-05456-f010:**
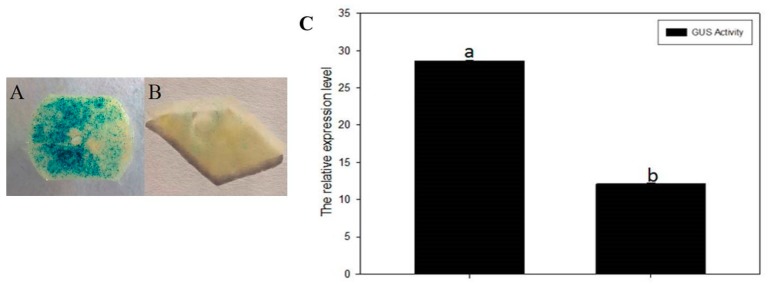
Tobacco transient expression to investigate the regulation of the *NtFLS* promoter by NtMYB3. (**A**) Tobacco leaf injected with pNtFLS::GUS; (**B**) Tobacco leaf injected with pNtFLS::GUS +*NtMYB3*; (**C**) QPCR analysis of GUS expression levels in tobacco leaves injected with pNtFLS::GUS (**1**) and injected with pNtFLS::GUS +*NtMYB3* (**2**). Small letters showed the significant difference at the level of *p* < 0.05 by using LSD test. (Scale bar = 1cm)

**Figure 11 ijms-20-05456-f011:**
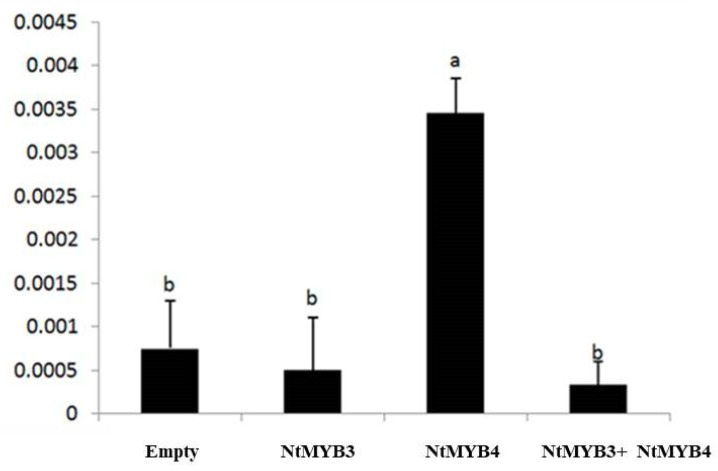
Dual luciferase assay confirms the repression of NtMYB3 to *NtFLS* promoter.

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
