# Peer review of "NtMYB3, an R2R3-MYB from Narcissus, Regulates Flavonoid Biosynthesis"

_ijms, 2019, doi:10.3390/ijms20215456_

Round 1
Reviewer 1 Report
The presented research is very complex and interesting, especially for horticulture, in order to bring some improvement in the colors of the ornamental flowers, such as Narcissus tazetta – Chinese narcissus. The obtained results could be used in this field and can represent an important scientific data for other similar studies.
A final conclusion of the research would be welcome to the reader.
Author Response
Dear Reviewer,
We appreciate you for spending time to review our paper and providing some valuable comments. we have uploaded the updated version of the manuscript.
Reviewer 2 Report
Anwar et al, isolated a novel R2R3 transcriptional factor from 13 Chinese narcissus. Transient assays showed that NtMYB3 reduced red pigmentation in agro-infiltrated leaves of tobacco while, the over-expression of NtMYB3 decreased the red color of transgenic tobacco flowers.The results of qPCR analysis indicated that over-expression of NtMYB3 strongly affected transcript level of different genes involved in flavonol, anthocyanin and proanthocyanin biosynthesis pathways. Regulation analysis of NtMYB3 to the NtFLS promoter was performed and results showed that the sequence of NtMYB3 has features of an R2R3-MYB repressor. The work is been write and discussed. Experimental procedure is satisfactory.
The work is ok in the present form. However, English must be revised.
Author Response
Response to Reviewer 2 Comments
Dear Reviewer, thank you very much for reviewing our manuscript. Your comments, kind suggestions and good recommendations are very useful for the improvement of the manuscript entitled “NtMYB3, an R2R3-MYB from narcissus regulate flavonoid biosynthesis”. We welcome further constructive comments if any.
Point 1: The work is ok in the present form. However, English must be revised.
Response 1: We have revised the manuscript and corrected the mistakes where need. We have improved the English of manuscript.
Reviewer 3 Report
Author documented this manuscript titled NtMYB3, an R2R3-MYB from Narcissus, Regulates Flavonoid Biosynthesis with detailed explanation of introduction, results and discussions. However, there are some additional data needed to support this manuscript. The comments are given below.
Author should go through the manuscript very carefully to fix some typos. For example, in abstract author started R2R3 MYB instead R2R3-MYB. Similar typos observed in the manuscript. Also, in line 77, author left double space between repress and anthocyanin.
Also, please check line 117 and 128. Author repeated the text two times.
Is NtMYB3 localized to nucleus?
Please explain transcript profiles of NtMYB3
Author Response
Response to Reviewer 3 Comments
Dear Reviewer,
Thank you very much for the review of our manuscript. Your comments, kind suggestions and good recommendations are very useful for the improvement of the manuscript. We have carefully considered the comments and tried our best efforts to address every one of them. We welcome further constructive comments if any.
Point 1: Author should go through the manuscript very carefully to fix some typos. For example, in abstract author started R2R3 MYB instead R2R3-MYB. Similar typos observed in the manuscript. Also, in line 77, author left double space between repress and anthocyanin.
Response 1: We have checked and revised through the whole manuscript. All the mistakes have been corrected according to your kind suggestions. In abstract R2R3 MYB has been replaced with R2R3-MYB. Please see line 12. We have removed the double space between repress and anthocyanin. Please see the line 77.
Point 2: Also, please check line 117 and 128. Author repeated the text two times.
Response 2: The repeated text in line 128 has been replaced with the text “Multiple alignments of the protein sequences of NtMYB3 with other R2R3-MYB repressors belonging to subgroup 4”.This information has been updated in line 128 and 129.
Point 3: Is NtMYB3 localized to nucleus?
Response 3: Yes, NtMYB3 localized to the nucleus. Please see the lines 97-100.
Point 4: Please explain transcript profiles of NtMYB3
Response 4: We have revised the transcript profiles of NtMYB3. Please see the lines 201-202 and 204-205.